# Preliminary Study on an Integrated System Composed of a Floating Offshore Wind Turbine and an Octagonal Fishing Cage

Chenglin Zhang [1,2], Jincheng Xu [1], Jianjun Shan [1], Andong Liu [1], Mingchao Cui [1], Huang Liu [1], Chongwu Guan [1] and Shuangyi Xie [3,*]

1    Fishery Machinery and Instrument Research Institute, Chinese Academy of Fishery Sciences, Shanghai 200092, China
2    College of Engineering Science and Technology, Shanghai Ocean University, Shanghai 201306, China
3    College of Mechanical Engineering, Chongqing University of Technology, Chongqing 400054, China
*    Correspondence: xsy1986@cqut.edu.cn

**Abstract:** To maximize the utilization of ocean resources, shorten the return period of investment and directly supply energy to the fishing cage, this paper performs a preliminary study for a state-of-the-art concept integrating a floating offshore wind turbine with a fishing cage. An octagonal semisubmersible rigid fishing cage with a slack catenary mooring system is designed to match the NREL 5 MW offshore baseline wind turbine. Combined with the blade pitch controller, fully coupled aero-hydro-elastic-servo-mooring simulations are performed through FAST and AQWA to explore the dynamic performance of the integrated system. Free decay conditions, uniform wind with irregular and regular waves, and turbulent wind with irregular waves are tested. The results showed that the integrated system works normally at the operating conditions and exhibits different dynamic characteristics for various scenarios. Additionally, the study on the influence of mooring line length indicates that the increasing line length can significantly affect the cage surge motion and the maximum and mean values of the upwind line tension at fairlead. Specifically, the maximum surge motion with a 924-m-long line is 404.8% larger than that with an 880-m-long line. When the line length increases by 5%, the maximum and mean line tensions decrease by 45.7% and 47.7%, respectively, while when the line length increases by 10%, the maximum and mean line tension decrease by 52.9% and 54.2%, respectively. It should be noted that the main purpose of this work is to conduct a preliminary study on this integrated system, aiming to provide an idea for the conceptual design, modeling and simulation analysis of this integrated system.

**Keywords:** fishing cage; floating offshore wind turbine; dynamics modeling; mooring system; net system





## 1. Introduction

With the continuous growth of the global population and worsening environment, land resources have been unable to meet the needs of human society. Additionally, the human demand for high-quality seafood is increasing strongly. The vast ocean with 360 million square kilometers is not only an important food source for human beings, but also a "blue granary" for getting high-end food and high-quality protein. Cage aquaculture is a method of cultivating aquatic products by placing a cage composed of a net, frame, buoyancy device, and fixing device in a specific sea area. This aquaculture method has been rapidly developed during the past decades [1] because of a series of advantages, such as high yield. However, nearshore fish aquaculture is facing more and more environmental problems, such as nearshore water pollution and occupation of nearshore space. In view of this, fish farm operators all over the world are considering relocating their farms to offshore locations, so as to make better use of continuous water flow and deep waters to disperse pollutants in a wider ocean space [2,3].

In recent years, some fish farming companies have put forward some designs of offshore fishing cages, and built them in selected offshore locations for testing [4]. Many ongoing projects use semisubmersible steel cages for offshore fish culture. Because the semisubmersible platform has been widely used in the oil and gas industry, it is natural to consider transforming it for marine fish farming. Ocean Farm 1, designed by Global Maritime, is a newly installed semi-submersible rigid cage (Figure 1). It is suitable for waters with water depths of 100 m to 300 m. It mainly consists of floating pontoons, slender frames and mooring systems [5]. However, it is not easy to carry out routine feeding and maintenance operations at offshore sites. Offshore fish farming, therefore, has to rely on remote technologies, such as unmanned surveillance and automated electrical equipment. These support devices require a constant power supply. Therefore, offshore fishing farms must have their own electricity supply, which can be derived from environmental power, such as solar, wind or wave energy [4].

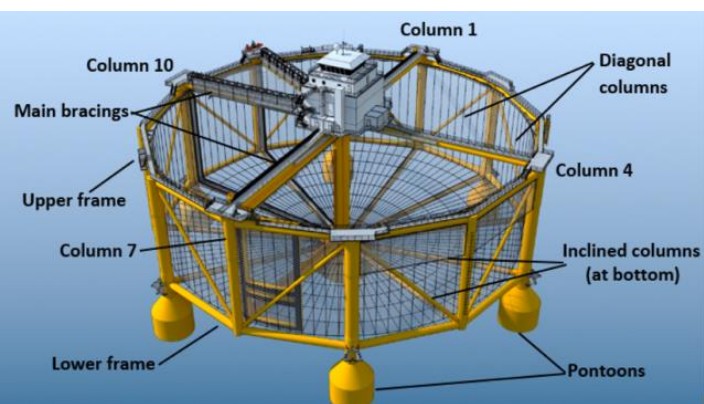

**Figure 1.** Illustration of Ocean Farm 1 [1]. Reprinted/adapted with permission from Ref. [1]. 2021, Jin, J.; Su, B.; Dou, R.; Luan, C.; Li, L.; Nygaard, I.; Fonseca, N.; Gao, Z.

In order to realize the sustainable development of human society, renewable energy is playing an increasingly important role in the total amount of social energy, with its advantages of large reserves, wide distribution and no pollution [6]. Among renewable energy sources, wind energy, especially offshore wind energy, is one of the most important renewable energy sources. It is considered to be a potential renewable energy resource to supplement traditional fossil fuels [7] and continues to grow rapidly around the world [8,9]. Wind turbines have become widely distributed due to advanced extraction technology. Compared with the onshore wind turbines, floating offshore wind turbines have a higher power generation efficiency due to the more abundant wind resources. Therefore, the development of the floating offshore wind turbines is considered a solution to deal with the energy crisis. Recently, research on simulations and experiments of floating offshore wind turbines have been widely carried out by many scholars [10–12]. For example, Russo et al. [11] presented new large-scale laboratory data on a physical model of a spar-type wind turbine with angular motion of control surfaces implemented. The experiments showed that the inclusion of pitch-controlled, variable-speed blades in physical tests on such types of structures is crucial. In the design of floating offshore wind turbines, in addition to considering the stability of wind turbine operation, the economy and investment return period also need to be considered emphatically.

In view of the above analyses, only a few studies have been performed integrating an aquaculture cage with a floating wind turbine, in recent years [2,13,14]. In Ref. [2], a 1 MW floating spar wind turbine and a fish cage is combined, named COSPAR. The COSPAR fish cage has four catenary mooring lines attached to the spar. Results showed that the COSPAR fish cage enhanced hydrodynamic responses compared with the floating fish cage with only four catenary lines connected to the side vertical columns of the cage. However, the influence of aerodynamic loads on the COSPAR fish cage is not considered. Ref. [13] also

proposed a state-of-the-art concept integrating a floating offshore wind turbine with a steel fishing cage, named FOWT-SFFC. The aero-hydro-servo-elastic modeling and time-domain simulations were performed using FAST to study the dynamic response of FOWT-SFFC. However, some simplifications were assumed, i.e., the drag force on the fish nets was neglected. Additionally, the mooring lines were modeled using the quasi-static method. Lei et al. [14] investigated the influence of nets on the dynamic response of a floating offshore wind turbine integrated with a steel fish cage. The results showed that nets play an important role in responses when wave periods are far away from natural periods of motion.

This novel concept of integrating a floating offshore wind turbine with a fishing cage can maximize the utilization of ocean resources, and it can be regarded as a reference for constructing a new pattern of offshore wind power integration development with harmonious coexistence between humans and nature. Therefore, this integrated system is worthy of further study. Based on this, a fully coupled aero-hydro-servo-elastic-mooring model of the integrated wind turbine–fishing cage system is established in this work. A series of simulations are carried out to explore the dynamic characteristics and feasibility of the integrated system. The structure of this paper is as follows.

In Section 2, the structural model, including the fishing cage, net, mooring lines, and wind turbine, is described. In Section 3, the dynamics model of the integrated system is built. To achieve coupled simulations, a control system is designed for the integrated system in Section 4. In Section 5, free decay tests, uniform wind with irregular and regular wave tests, turbulent wind and irregular wave tests are performed. In addition, the influence of mooring line length are also investigated. Finally, the conclusions are provided in Section 6.

## 2. Structural Model

The integrated system comprises two parts: (1) a semisubmersible fishing cage with a slack catenary mooring system, and (2) a wind turbine on the top of the cage to provide electrical power, as shown in Figure 2. The concept of the integrated system basically aims at supporting offshore fish culture and harvesting wind energy at the same time to provide a power supply for the offshore fish culture.

### 2.1. Frame Structure of Fishing Cage

In this paper, in addition to the fish culture function, the fishing cage also serves as a floating foundation to support the upper wind turbine. The frame structure of the fishing cage consists of vertical columns (inner and outside, nine in total), heave plates (attached to the bottom of vertical columns, five in total), horizontal girders (top and bottom, 16 in total), X-shaped girders (sides, eight in total), radial girders (top sides and bottom sides, four and eight in total, respectively). The inner space of the fishing cage can be subdivided into four sectors to cultivate different kinds of fish. The overall shape of the fishing cage is similar to Ocean Farm 1 [1], which is the reference fishing cage design. The upper wind turbine can be installed on the top of the central column. Figure 3 shows views of the fishing cage with key dimensions.

The total draft of the fishing cage is 41 m, while the deepest culture depth in the fishing cage is 35 m, considering that a deeper water site may affect the fish growth due to the lack of light and oxygen saturation, or the significant change in water temperature. Thus, the total height of the fishing cage is 51 m (10 m above water and 41 m underwater). The inner water holding space of the fishing cage is approximately 120,000 m$^3$.

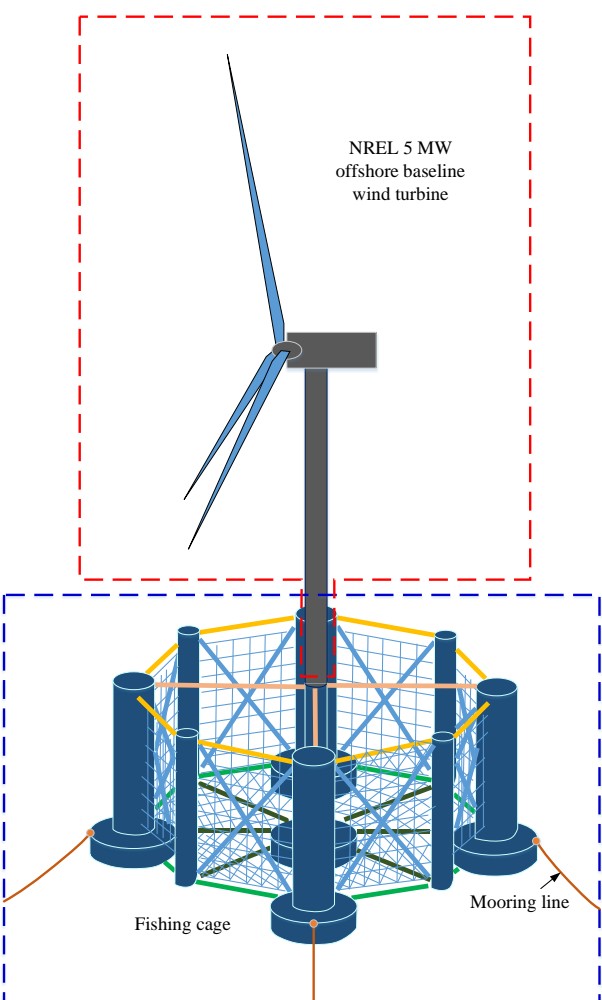

**Figure 2.** Schematic diagram of the integrated system of a floating offshore wind turbine and a fishing cage.

In order to match the properties of the upper wind turbine tower, the central column has a diameter of 6.5 m, and the elevation of the fishing cage above the still water level is set to 10 m. There are altogether four vertical columns with a diameter of 12 m and a thickness of 0.06 m. There are also four vertical columns with a diameter of 6 m and a thickness of 0.03 m. Seen from the top view, the eight vertical columns are evenly arranged and connected into a regular octagon by horizontal girders. The outside columns are connected to the central column by four radial girders at the top and eight radial girders at the bottom so that the whole fishing cage works as a rigid body. The radial girders have the same diameter (1.6 m) and thickness (0.0175 m). Each column has a heave plate to suppress the heave motion of the fishing cage. The height, diameter and thickness of the heave plates are 6, 24 and 0.06 m, respectively. The above dimensions of the fishing cage are obtained taking into account the hydrostatic balance of the cage in the still water condition and the sufficient amount of water for fish culture, in addition to referring to the dimensions of Ocean Farm 1 [1] and OC4-DeepCwind platform [15]. Table 1 summarizes the key dimensions of the structural members of the fishing cage.

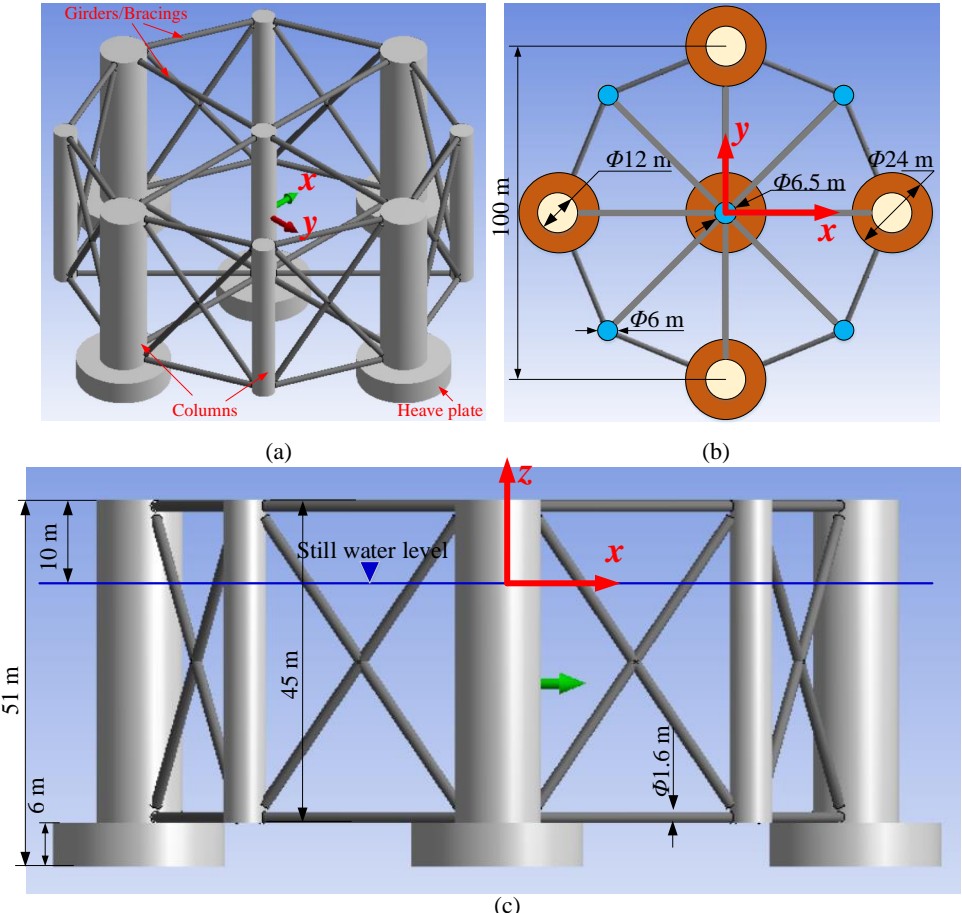

**Figure 3.** View of the fishing cage with key dimensions (without nets). (**a**) ISO view; (**b**) Top view; (**c**) Side view.

**Table 1.** Geometric properties of the fishing cage.

| Property | Value |
|---|---|
| Total draft | 41 m |
| Elevation of central column above the still water level | 10 m |
| Elevation of outside columns above the still water level | 10 m |
| Distance from central column centerline to outside column centerline | 50 m |
| Height of central and outside columns | 45 m |
| Height of heave plates | 6 m |
| Diameter of each outside column with a heave plate | 12 m |
| Diameter of each outside column without a heave plate | 6 m |
| Diameter of heave plates | 24 m |
| Diameter of central column | 6.5 m |
| Diameter of bracings | 1.6 m |
| Thickness of each outside column with a heave plate | 0.06 m |
| Thickness of each outside column without a heave plate | 0.03 m |
| Thickness of heave plates | 0.06 m |
| Thickness of central column | 0.03 |
| Thickness of bracings | 0.0175 m |

The total mass of the fishing cage, including ballast, is calculated such that the combined weight of the rotor-nacelle assembly, tower and fishing cage, plus the weight of the mooring system in water should balance with the buoyancy of the undisplaced cage in still water. The concrete, with a high density of 2,400 kg/m³, is used in heave plates for ballast, which is added from the bottom of heave plates upwards until the mass requirement is met.

The material of the fishing cage is selected as steel with an effective density of 8,500 kg/m³. The masses of the frame structure, girders and ballast are 8,264,815, 1,304,543 and 27,449,312 kg, respectively. The center of mass of the whole fishing cage is located at about 34 m below the still water level. Table 2 summarizes the structural properties of the undisplaced cage.

**Table 2.** Structural properties of the fishing cage.

| Property | Value |
|---|---|
| Frame structure, bracing, ballast mass | $8.265 \times 10^6$, $1.305 \times 10^6$, $2.745 \times 10^8$ kg |
| Center of mass location below the still water level, including ballast and bracings | $-34.07$ m |
| Pitch inertia about the center of mass, including ballast and without bracings | $4.1867 \times 10^{11}$ kg m² |
| Roll inertia about the center of mass, including ballast and without bracings | $4.1867 \times 10^{10}$ kg m² |
| Yaw inertia about the center of mass, including ballast and without bracings | $7.5657 \times 10^{10}$ kg m² |

### 2.2. Net System

The net is stretched over the sides and bottom of the fishing cage. The type of net used in Ocean Farm 1 is EcoNet, which is a non-fiber material with a hard surface that can resist ocean pollution [5]. It is made from very strong but light PET (Polyethylene Terephthalate), and has been certified according to the Norwegian fish farming standard NS 9415 [16], and its service lifetime in the water can reach up to 14 years.

The solidity ratio is the ratio of the projected area of screen threads to the total area of the net panel. For a square net, as illustrated in Figure 4, the solidity ratio $S_n$ is calculated by [5,17]:

$$S_n = \frac{2d_w}{l_w} - \left(\frac{d_w}{l_w}\right)^2 \tag{1}$$

where $d_w$ is the diameter of twine, $l_w$ is the length of twine.

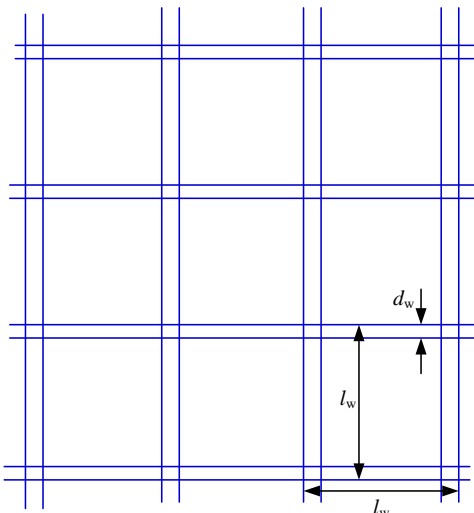

**Figure 4.** Illustration of net wine.

In the analysis, it is necessary to consider the wave forces applied to the net attached to the fishing cage and the total force for all net elements. Nevertheless, the number of all cells in a fish net is too large to be included in a model. In order to improve simulation efficiency, it is necessary to put forward a simplified approach, so that the total net force can be transferred to the rigid cage. Therefore, an equivalent net method, which has been studied by Duo [5] and Li and Ong [18], is used in this paper. The solidity ratio of the net

on Ocean Farm 1, 0.16 [19], is applied in this work. An equivalent diameter $d_w$ = 0.4 m is introduced, with an assumed twine length $l_w$ = 4.8 m, to obtain the same solidity ratio based on Equation (1).

### 2.3. Mooring Line System

A uniformly distributed four-line spread mooring system is applied for the integrated system. The fairleads are located at the outside heave plates, at a depth of 35 m below the sea water level and at a radius of 62 m from the cage center. The anchors are located at a water depth of 200 m and at a radius of 891.6 m from the cage center. The mooring lines are made from a studless grade 4 chain, with a chain nominal diameter of 0.153 m [19]. The unstretched length of each mooring line is 880 m, a submerged unit mass of 447 kg/m, an axial stiffness of $2.1 \times 10^6$ kN and a catalogue breaking strength of $2 \times 10^4$ kN [19]. Note that the length of 880 m is determined after some study on the mooring system of the DeepCWind semisubmersible platform at the same water depth [20]. All hydrodynamic coefficients are determined according to the chain's nominal diameter. The drag and the added mass coefficients are assumed based on DNVOS-E301 [21]. Tables 3 and 4 list the configurations and properties of the mooring line system, respectively. Figure 5 shows the mooring line system configuration.

**Table 3.** Configurations of mooring line system.

| Configuration | Value |
|---|---|
| Number of mooring lines | 4 |
| Angle between adjacent lines | 90° |
| Water depth | 200 m |
| Depth from fairlead to still water level | 35 m |
| Radius from anchors to cage centerline | 891.6 m |
| Radius from fairleads to cage centerline | 62 m |
| Unstretched mooring line length | 880 m |

**Table 4.** Properties of mooring line system.

| Property | Value |
|---|---|
| Chain type | Studless grade 4 |
| Nominal diameter | 0.153 m |
| Unit mass in water | 447 kg/m |
| Axial stiffness | $2.1 \times 10^6$ kN |
| Catalogue breaking strength | $2 \times 10^4$ kN |
| Transversal drag coefficient | 2.4 |
| Longitudinal drag coefficient | 1.15 |
| Added mass coefficient | 1 |

### 2.4. Wind Turbine System

In this work, the upper wind turbine of the integrated FOWTs is selected as the NREL offshore 5 MW baseline wind turbine [22]. For the convenience of understanding, the properties of the wind turbine will be described briefly in this section. More details can be found in Ref. [22].

To support the preliminary study aimed at assessing the integrated system composed of a floating offshore wind turbine and a fishing cage, the NREL offshore 5 MW baseline wind turbine, a representative utility-scale multi-megawatt turbine, is applied in this work. This wind turbine is a conventional three-bladed upwind variable-speed variable-pitch turbine. Table 5 shows the main properties of the NREL offshore 5 MW baseline wind turbine. More details can be referred to Refs. [22,23].

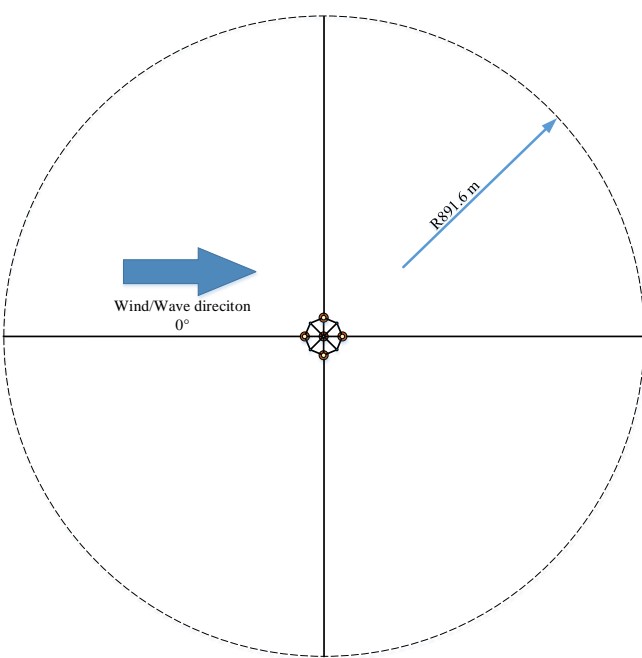

**Figure 5.** Top view of the mooring line system configuration.

**Table 5.** Main properties of the NREL offshore 5 MW wind turbine.

| Property | Parameter |
|---|---|
| Rating power | 5 MW |
| Rotor orientation, Configuration | Upwind, Three Blades |
| Control | Variable Speed, Collective Pitch |
| Drivetrain | High Speed, Multiple-Stage Gearbox |
| Rotor, Hub diameter | 126 m, 3 m |
| Hub height | 90 m |
| Cut-In, Rated, Cut-Out wind speed | 3, 11.4, 25 m/s |
| Cut-in, Rated rotor speed | 6.9 rpm, 12.1 rpm |
| Rated tip speed | 80 m/s |
| Overhang, Shaft tilt, Precone | 5 m, 5°, 2.5° |
| Rotor, Nacelle mass | 110,000 kg, 240,000 kg |

The tower base is coincident with the top of the central column of the fishing cage and is located at an elevation of 10 m above the still water level. The tower top is coincident with the yaw bearing and is located at an elevation of 87.6 m above the still water level. This tower-top elevation (90 m above the still water level) is consistent with the land-based version of the NREL 5 MW baseline wind turbine, as described in Ref. [23]. These properties are all relative to the undisplaced position of the fishing cage.

The diameter at the tower base for the NREL 5-MW offshore baseline wind turbine is 6.5 m, which matches the diameter of the central column of the fishing cage. The tower–base thickness, top diameter and thickness are 0.027, 3.87 and 0.019 m, respectively. The effective mechanical steel properties of the tower are determined according to the DOWEC study [24]. Young's modulus and shear modulus are set to 210 GPa and 80.8 GPa, respectively. The effective density of the steel is set as 8500 kg/m$^3$, which is meant to be an increase above steel's typical value of 7850 kg/m$^3$, to consider paint, bolts, welds and flanges that are not included in the tower thickness data [15]. The tower radius and thickness are assumed to be linearly tapered from the tower base to the tower top.

The overall tower mass is 249,718 kg and is centered at 43.4 m along the tower centerline above the still water level. This is obtained from the overall tower length of 77.6 m. A structural damping ratio of 1% critical is specified for all modes of the isolated

tower, which corresponds to the values used in the DOWEC study [24]. Table 6 gives the undistributed tower properties.

**Table 6.** Undistributed tower properties.

| Property | Parameter |
|---|---|
| Elevation to tower base above still water level | 10 m |
| Elevation to tower top above still water level | 87.6 |
| Overall tower mass | 249,718 kg |
| Center of mass location above still water level | 43.4 m |
| Structural damping ration | 1% |

## 3. Dynamics Modeling

In the present work, the fully coupled aero-hydro-elastic-servo-mooring model of the integrated FOWT is established through a coupling framework based on FAST [25] and ANSYS/AQWA [26]. The upper wind turbine is modeled on FAST. The turbine dynamic responses are represented by external forces and added mass within each invocation, then are passed to the AQWA solver to be combined with the hydrodynamic loads and mooring line forces to calculate the dynamic responses of the aquaculture cage. For the convenience of understanding, the subsequent sections will briefly describe the basic theories applied in FAST and AQWA for structural modeling. Ref. [27] gives more implementation details.

Figure 6 shows the modeling overview of the integrated system through FAST and AQWA in this work.

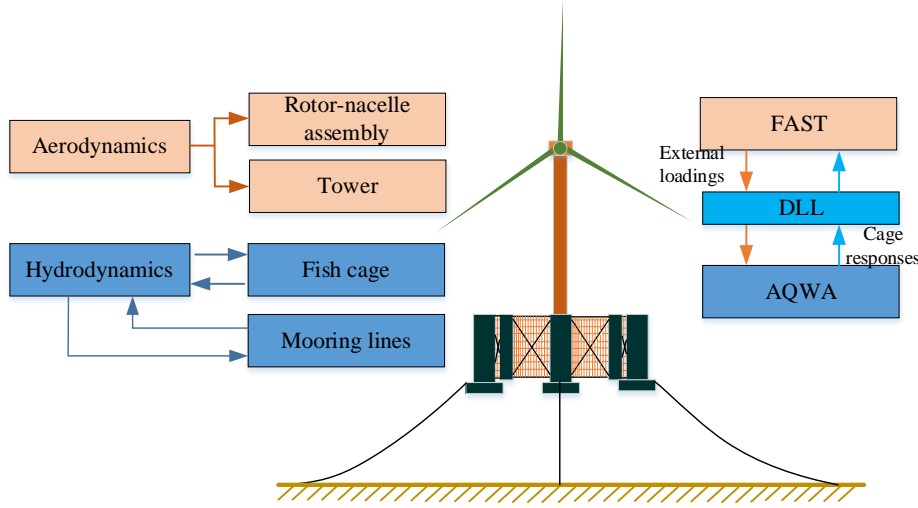

**Figure 6.** Modeling overview through FAST and AQWA.

### 3.1. Aerodynamics and Structural Dynamics

FAST is an aero-hydro-elastic-servo wind turbine simulation tool developed by the National Renewable Energy Laboratory (NREL) [25]. The aerodynamic loads on the blades are calculated by the blade element momentum (BEM) theory within the AeroDyn module [28,29]. The rotor thrust ($T$) and torque ($M$) are calculated as:

$$\begin{cases} \mathrm{d}T = \frac{1}{2}\rho_{\mathrm{air}}W^2c(C_l\cos\varphi + C_d\sin\varphi)\mathrm{d}r \\ \mathrm{d}M = \frac{1}{2}\rho_{\mathrm{air}}W^2c(C_l\sin\varphi - C_d\cos\varphi)r\mathrm{d}r \end{cases} \tag{2}$$

where $\rho_{\mathrm{air}}$ is the air density, $C_l$ and $C_d$ are, respectively, the lift and drag coefficients of the airfoil, $W$ is the absolute wind speed, $c$ is the chord length of a blade element, $\varphi$ is the inflow angle and $r$ is the local radius of a blade element.

More detailed information on the aerodynamic calculation can be found in Refs. [28,29]. When the aerodynamic loads are calculated, they are then imported into the electrodynamic

module of FAST to resolve the motion equations of the wind turbine. Kane's kinetic approach is used, which is defined as

$$\mathbf{F}_r^* + \mathbf{F}_r = 0 \tag{3}$$

where $\mathbf{F}_r^*$ and $\mathbf{F}_r$ are the generalized inertia force vector and the generalized active force vector, respectively.

The generalized inertia forces are composed of the inertia forces of the nacelle, tower, hub and blades. The generalized active forces consist of aerodynamic loads $\mathbf{F}_{aero}$, elastic restoring forces $\mathbf{F}_{elas}$, gravity $\mathbf{F}_{grav}$ and damping forces:

$$\mathbf{F}_r = \mathbf{F}_{aero} + \mathbf{F}_{elas} + \mathbf{F}_{grav} + \mathbf{F}_{damp} \tag{4}$$

The wind turbine is modelled as a multi-body system including rigid bodies and flexible bodies. The hub and nacelle are modelled as rigid bodies. The tower and blades are treated as flexible bodies. The generalized inertia force of a rigid body is represented by the same formula. Further information on the motion equations can be referred to in Refs. [30,31].

### 3.2. Mooring Line Dynamics

The finite element approach within AQWA is applied to consider the dynamic effects of mooring lines. Each line is discretized into several finite elements, and the mass of each element is concentrated into a corresponding node, as shown in Figure 7. In the figure, $S_j$ is the length of an unstretched line between the anchor and the $j$th node, and $D_e$ represents the local segment diameter of one line. Each line is treated as a chain of Morison elements subjected to various external forces. According to Ref. [32], the motion equation of each line element is defined as:

$$\begin{cases} \frac{\partial \mathbf{T}}{\partial S_e} + \frac{\partial \mathbf{V}}{\partial S_e} + \mathbf{w} + \mathbf{F}_h = m_e \frac{\partial^2 \mathbf{R}}{\partial t^2} \\ \frac{\partial \mathbf{M}}{\partial S_e} + \frac{\partial \mathbf{R}}{\partial S_e} \times \mathbf{V} = -\mathbf{q} \end{cases} \tag{5}$$

where $\mathbf{T}$ and $\mathbf{V}$ are the tension force and shear force vectors of the first element node, respectively; $\mathbf{R}$ is the position vector of the first element node; $S_e$ is the length of an unstretched element; $\mathbf{w}$ and $\mathbf{F}_h$ represent the weight and hydrodynamic load vectors per unit element length, respectively; $m_e$ is the mass per unit length; $\mathbf{M}$ is the bending moment vector of the first element node; and $\mathbf{q}$ is the distributed moment load per unit length.

The bending moment and tension force vectors are calculated by:

$$\begin{cases} \mathbf{M} = EI \cdot \frac{\partial \mathbf{R}}{\partial S_e} \times \frac{\partial^2 \mathbf{R}}{\partial S_e^2} \\ \mathbf{T} = EA \cdot \varepsilon \end{cases} \tag{6}$$

where $EI$ and $EA$ are, respectively, the bending stiffness and axial stiffness of the line, and $\varepsilon$ is the strain of the line.

To ensure a unique solution to Equation (5), pinned connection boundary constraints (Equation (7)) are imposed on the top and bottom ends:

$$\begin{cases} \mathbf{R}(0) = \mathbf{P}_{bot}, \mathbf{R}(L) = \mathbf{P}_{top} \\ \frac{\partial^2 \mathbf{R}(0)}{\partial S_e^2} = 0, \frac{\partial^2 \mathbf{R}(L)}{\partial S_e^2} = 0 \end{cases} \tag{7}$$

where $\mathbf{P}_{bot}$ and $\mathbf{P}_{top}$ are the position vectors of attachment points, and $L$ is the length of the unstretched line.

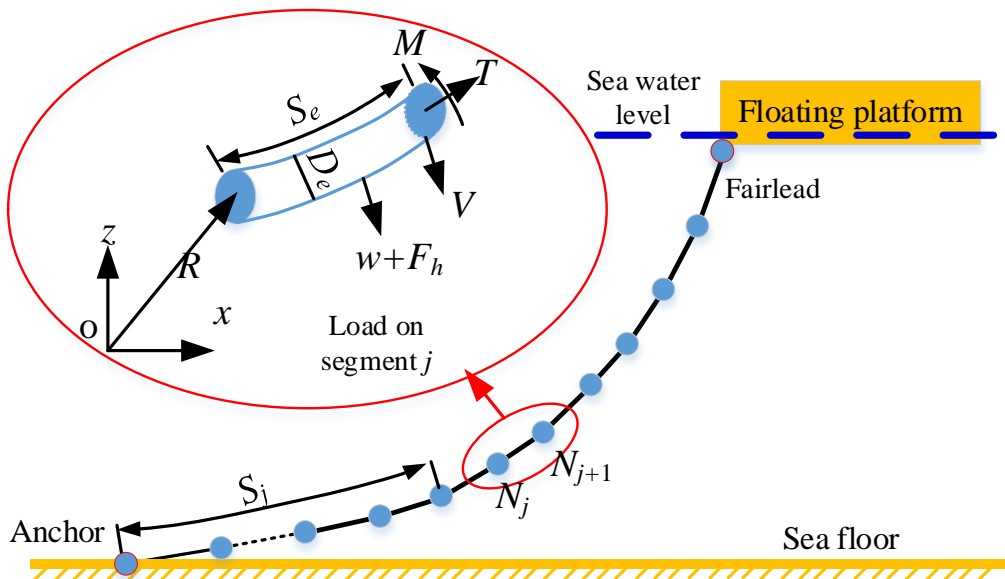

**Figure 7.** Schematic diagram of the dynamic model of mooring lines.

### 3.3. Hydrodynamics of Aquaculture Cage

The hydrodynamic loads on the aquaculture cage are obtained based on the frequency-dependent hydrodynamic coefficients, including the added mass, radiation damping and mooring line restoring forces. These coefficients can be obtained through a frequency domain analysis in AQWA.

### 3.3.1. Hydrodynamics of Cage Support Structures

The hydrostatic and hydrodynamic analyses are performed by the panel method to solve the radiation and diffraction problems for the interaction of surface waves with main structures in the frequency domain. Based on boundary conditions, the velocity potentials can be solved on the mean body position, and the pressure can be obtained from the linear Bernoulli's equation [2]. The governing equation for the motion of a floating body in six degrees of freedom (DOFs) is expressed by [2]:

$$\left( -\omega^2 (M + A(\omega)) + i\omega B(\omega) + C \right) \zeta(\omega) = F(\omega) \tag{8}$$

where $M$ is the mass matrix, $\omega$ is the angular frequency, $A(\omega)$ is the added mass matrix, $B(\omega)$ is the damping matrix, $C$ is the hydrostatic stiffness matrix, $\zeta(\omega)$ is the dynamic response vector and $F(\omega)$ is the dynamic load vector.

When using the potential flow theory in AQWA, the diffraction panel elements model does not include the effect of the viscous drag force. Instead, the Morison elements are usually applied for slender structures. Therefore, the cross-bracings that act as a support role are modeled through a series of Morison elements. According to the setting in Ref. [15], the drag coefficient for the cross-bracings in this work is set to 0.63. Figure 8 shows the panel model of the aquaculture cage modeled in AQWA.

### 3.3.2. Net Hydrodynamics

In addition to the cage support structures, the nets of the fishing cage also need to be properly modeled. The nets are flexible and undergo various degrees of deformation depending on the wave and current conditions. The hydrodynamic force on nets and the coupling effect between nets and cage support structures have strong nonlinearity. Thus, a nonlinear model is usually needed to model the nets, and examples of flexible nets modeling can be found in Refs. [33–35].

Compared with the traditional flexible collar fishing cage, this semisubmersible rigid fishing cage has rigidity and larger supporting structures, so it is expected that the nets

have less influence on the fishing cage. Therefore, in this preliminary study, the equivalent nets are modeled by Morison elements, which are considered to be stiff and rigidly connected to the cage to work as a whole rigid body [14,18]. Thus, the relative motions between the nets and the cage are neglected. It should be noted that this simplification may overestimate the hydrodynamic forces and viscous effects of the nets because deformation is not considered [18]. Nevertheless, it is still a good start to study the global dynamic responses of the integrated system with the inclusion of the nets modeled by the simplified rigid model.

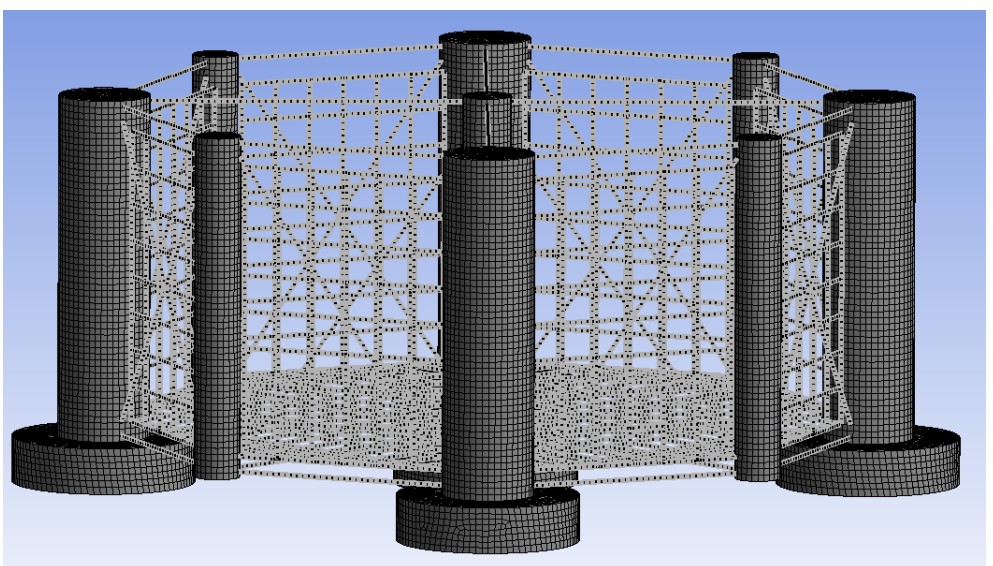

**Figure 8.** Panel model of the aquaculture cage in AQWA.

Hydrodynamic forces on the nets may be divided into three components: inertia force, drag force and lift force. The inertia force and drag force for the nets can be calculated by using the Morison equation. The lift force can be calculated by the same drag force term in the Morison equation, but the lift coefficient, instead of the drag coefficient, must be introduced [36]. Considering that the mass of the nets accounts for a small proportion of the whole cage mass, the inertia force of the nets may be ignored [5,37,38]. Moreover, when the nets are attached to the cage sides and bottom, the wave and flow directions are almost $0°$ or $90°$ to the normal direction of the net plane where the lift force is almost zero [15,39]. Therefore, in this paper, only the drag force on the nets is considered.

The hydrodynamic drag force of the nets is closely related to the net solidity ratio. In Refs. [36,40], a drag coefficient of nets, $C_D$, was estimated using the following equation:

$$C_D = 0.04 + (-0.04 + 0.33S_n + 6.54S_n^2 - 4.88S_n^3) \cos\theta \tag{9}$$

where $S_n$ is the net solidity ratio, $\theta$ is the angle between the inflow direction and the net normal.

In this work, it is assumed that the angle of inflow to the net normal is $0°$ for both wave inflow to the side nets and vertical motion inflow to the bottom nets. The drag coefficient $C_D = 0.2$ is determined according to the selected solidity ratio, which is applied for the equivalent nets at both the side and bottom. Note that the possible influence of biofouling on the drag coefficient is neglected in this work, which requires more accurate biofouling hydrodynamic characteristics. This issue needs to be solved by performing experiments in the future.

## 4. Control System

As studied by Jonkman [22], the operation regions of a wind turbine are divided into five parts: 1, 1.5, 2, 2.5, and 3, as illustrated in Figure 9. In Region 1, the wind speed

is lower than the cut-in wind speed, thus, no electrical power is output. In this region, the rotor is accelerating for a start-up. Region 1.5 is a linear transition between Region 1 and Region 2. In Region 2, the controller adjusts the generator torque according to the generator speed while keeping the blade pitch angles at the optimal value. In Region 3, the blade pitch angles are tuned by a collective variable-pitch strategy to maintain the rated generator speed. The demanded blade pitch angles are provided through a gain-scheduled proportional-integral (PI) controller, depending on the speed error between the filtered and the rated generator speeds. Moreover, in order to resist negative damping in the rotor speed response, the generator-torque control law in Region 3 is set to a constant generator-torque control. The constant generator torque is set to the rated value, 43,093.55 Nm [23]. Region 2.5 is a smooth transition region between Regions 2 and 3. This region is also applied to limit tip speed and noise emissions.

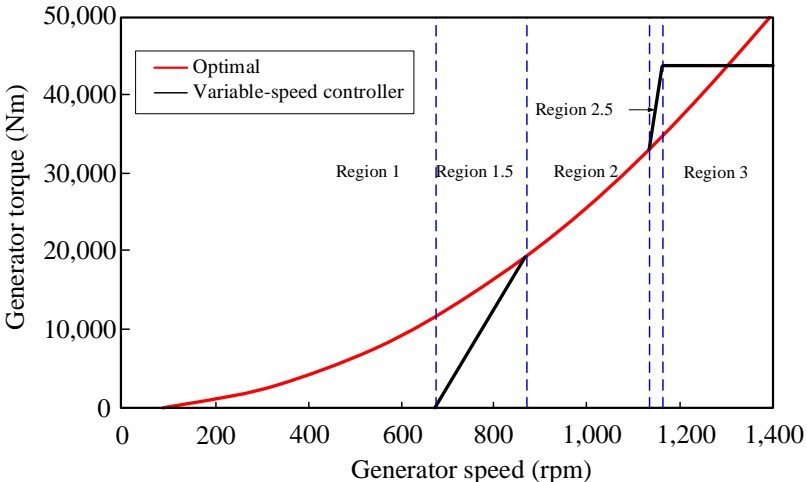

**Figure 9.** Relationship between generator torque and generator speed for the baseline control system.

The blade pitch control system can be represented by the following equation of motion [41]:

$$\underbrace{\left[I_{\text{drivetrain}} + \frac{1}{\Omega_0}\left(-\frac{\partial P}{\partial\theta}\right)N_{\text{gear}}K_{\text{D}}\right]}_{M_\phi}\ddot{\varphi} + \underbrace{\left[\frac{1}{\Omega_0}(-\frac{\partial P}{\partial\theta})N_{\text{gear}}K_{\text{P}} - \frac{P_0}{\Omega_0^2}\right]}_{C_\phi}\dot{\varphi} + \underbrace{\left[\frac{1}{\Omega_0}(-\frac{\partial P}{\partial\theta})N_{\text{gear}}K_{\text{I}}\right]}_{K_\phi}\varphi = 0 \quad (10)$$

where $I_{\text{drivetrain}}$ is the drivetrain inertia cast to the low-speed shaft; $N_{\text{gear}}$ is the gearbox ratio; $\Omega_0$ is the rated rotor rotational speed; $P_0$ is the rated mechanical power; $\partial P/\partial\theta$ is the sensitivity of aerodynamic power to the rotor collective blade pitch angle; $K_{\text{P}}$, $K_{\text{I}}$ and $K_{\text{D}}$ are the blade pitch controller proportional, integral and derivative gains, respectively; $\dot{\varphi} = \Delta\Omega$ is the rotor speed error.

The rotor speed error responds as a 1-DOF dynamic system with natural frequency $\omega_{\varphi\text{n}}$ and damping ratio $\zeta_\varphi$:

$$\begin{cases} \omega_{\varphi\text{n}} = \sqrt{\dfrac{K_\varphi}{M_\varphi}} \\ \zeta_\varphi = \dfrac{C_\varphi}{2M_\varphi\omega_{\varphi\text{n}}} \end{cases} \quad (11)$$

when designing a blade pitch controller, the PI gains can be calculated by ignoring the derivative gain and negative damping term [42]:

$$\begin{cases} K_{\text{P}} = \dfrac{2I_{\text{drivetrain}}\Omega_0\zeta_\varphi\omega_{\varphi\text{n}}}{N_{\text{gear}}\left(-\frac{\partial P}{\partial\theta}\right)} \\ K_{\text{I}} = \dfrac{I_{\text{drivetrain}}\Omega_0\omega_{\varphi\text{n}}^2}{N_{\text{gear}}\left(-\frac{\partial P}{\partial\theta}\right)} \end{cases} \quad (12)$$

According to the study by Larsen [43], the smallest natural frequency of the blade pitch controller must be less than the smallest critical natural frequency of the support structure to ensure that the support structure motions of a floating offshore wind turbine with active pitch-to-feather control remain positively damped. Therefore, the blade pitch controller's natural frequency of 0.032 Hz (which is below the cage-pitch natural frequency of about 0.06 Hz) and a damping ratio of 0.7 [23] is used in this paper. The resulting proportional gain and integral gain are 0.006275604 s and 0.0008965149, respectively.

## 5. Results and Discussions

### 5.1. Free Decay Test

Free decay tests in six degrees of freedom (DOFs) are performed to determine the natural frequencies of the integrated system. The test results are shown in Figure 10 and the natural frequencies are listed in Table 7.

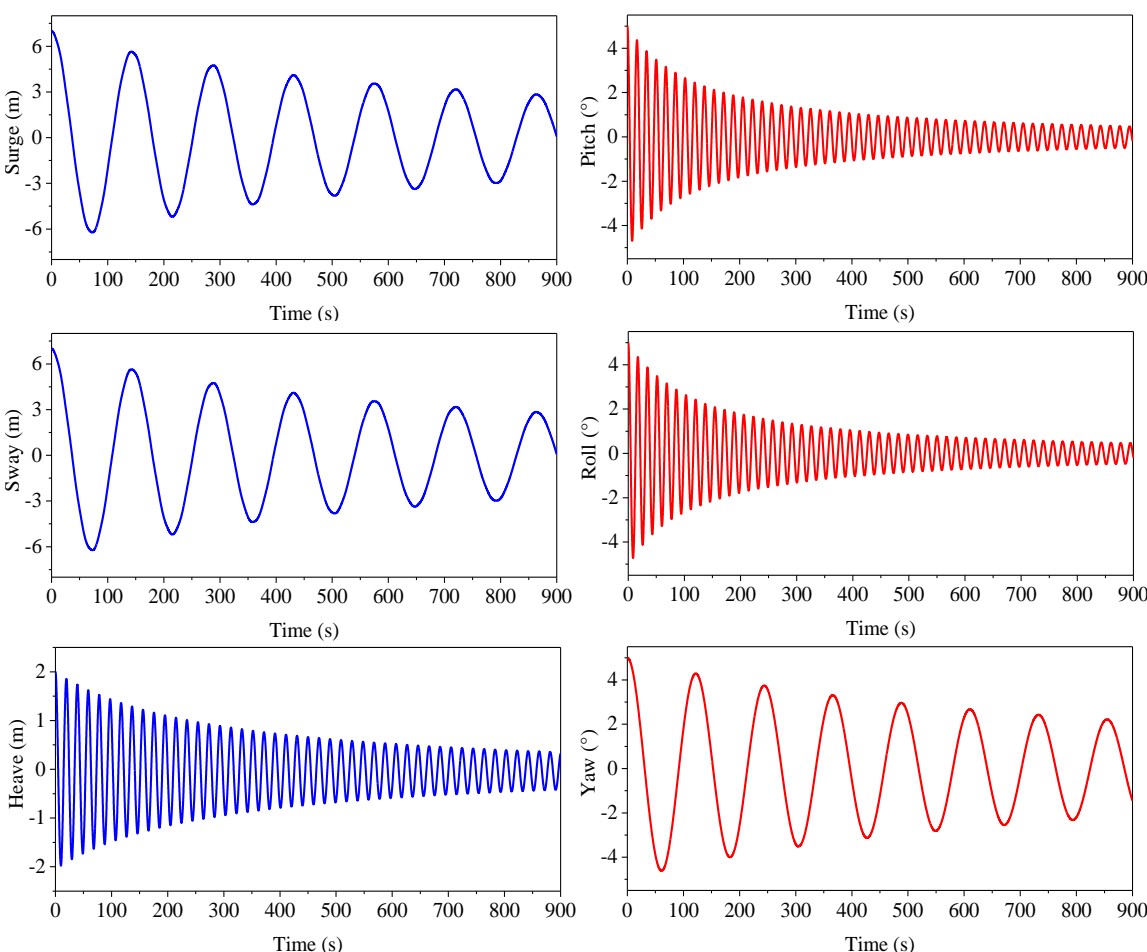

**Figure 10.** Free decay motions of the integrated system in six DOFs.

**Table 7.** Natural frequencies of the integrated system in six DOFs.

| DOF | Natural Frequency (Hz) |
| --- | --- |
| Surge | 0.00667 |
| Sway | 0.00667 |
| Heave | 0.05111 |
| Pitch | 0.05778 |
| Roll | 0.05778 |
| Yaw | 0.00778 |

It should be noted that because of the coupling of different DOFs, the integrated system will always generate some motions in other DOFs when performing free decay tests, especially the surge-pitch DOFs and sway-roll DOFs. For the free decay in the surge DOF (the initial surge displacement is set to 7 m), the largest pitch amplitude is almost 0.2°, while for the free decay in the pitch DOF (the initial pitch displacement is set to 5°), the largest surge amplitude is almost 3 m. Similar results are observed for the free decay tests in the sway and roll DOFs. These values are not so large when compared to those in the DOFs that are being tested. Thus, the results obtained from the free decay tests are considered reasonable.

## 5.2. Uniform Wind with Regular and Irregular Waves Test

The uniform wind with regular waves is firstly tested. For the uniform wind, a constant wind speed of 18 m/s, without considering wind shear, is applied. The regular waves are generated using the Airy wave theory. The significant wave height and peak period are set as 4.1 m and 10.5 s, respectively [19]. The simulation time is 700 s, and the results corresponding to the first 100 s are omitted. Figure 11 shows the time series of cage motions in six DOFs under the uniform wind with regular waves condition. It is found that the surge and have motions are more significant than the sway motion. For the rotational motions, the roll and yaw motions are close to zero. With the main action of aerodynamic loads, a mean pitch rotation of 0.48° is induced.

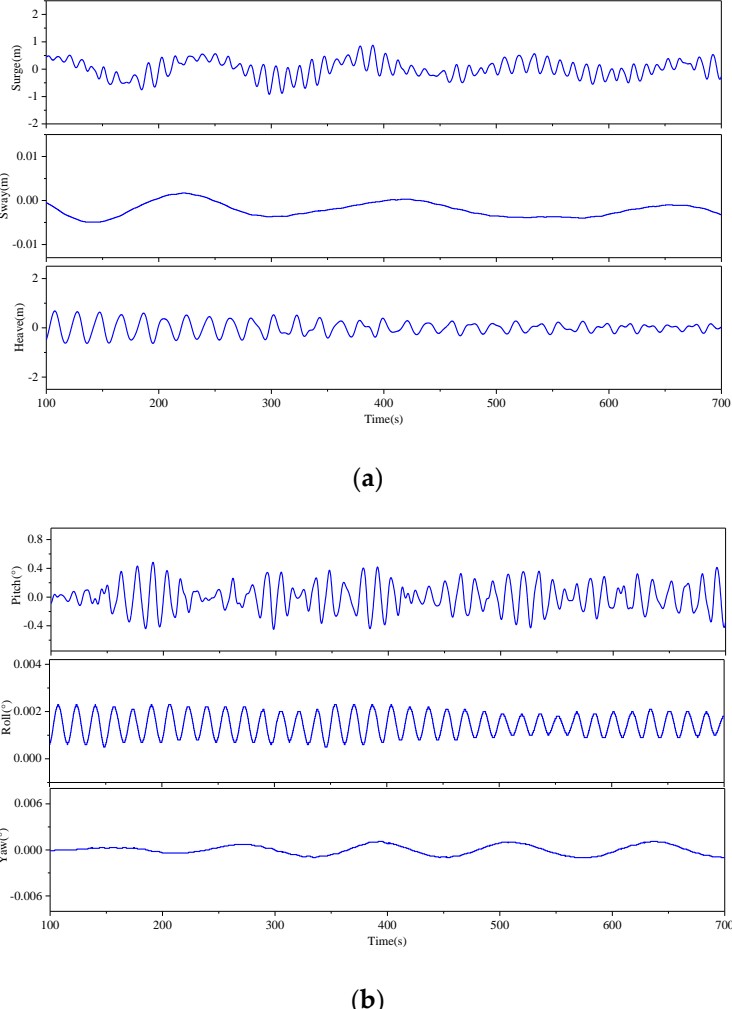

(**a**)

(**b**)

**Figure 11.** Time histories of cage motions under the uniform wind with regular waves. (**a**) Translational motion. (**b**) Rotational motion.

For the irregular waves, the sea state is generated by the JONSWAP spectrum. Figure 12 shows the time series of cage motions in six DOFs under the uniform wind with irregular waves condition. For the translational motions, the surge motion is more significant than the heave and sway motions. A mean surge motion of approximately 2.7 m is mainly caused by the aerodynamic loads. For the rotational motions, the roll and yaw motions are close to zero. With the main action of aerodynamic loads, a mean pitch rotation of 0.18° is induced. Additionally, it is expected that no cage-pitch resonance is found because the blade pitch controller's natural frequency is below the cage-pitch natural frequency. Overall, it is observed that, compared with the regular wave condition, the irregular condition causes more significant cage motions.

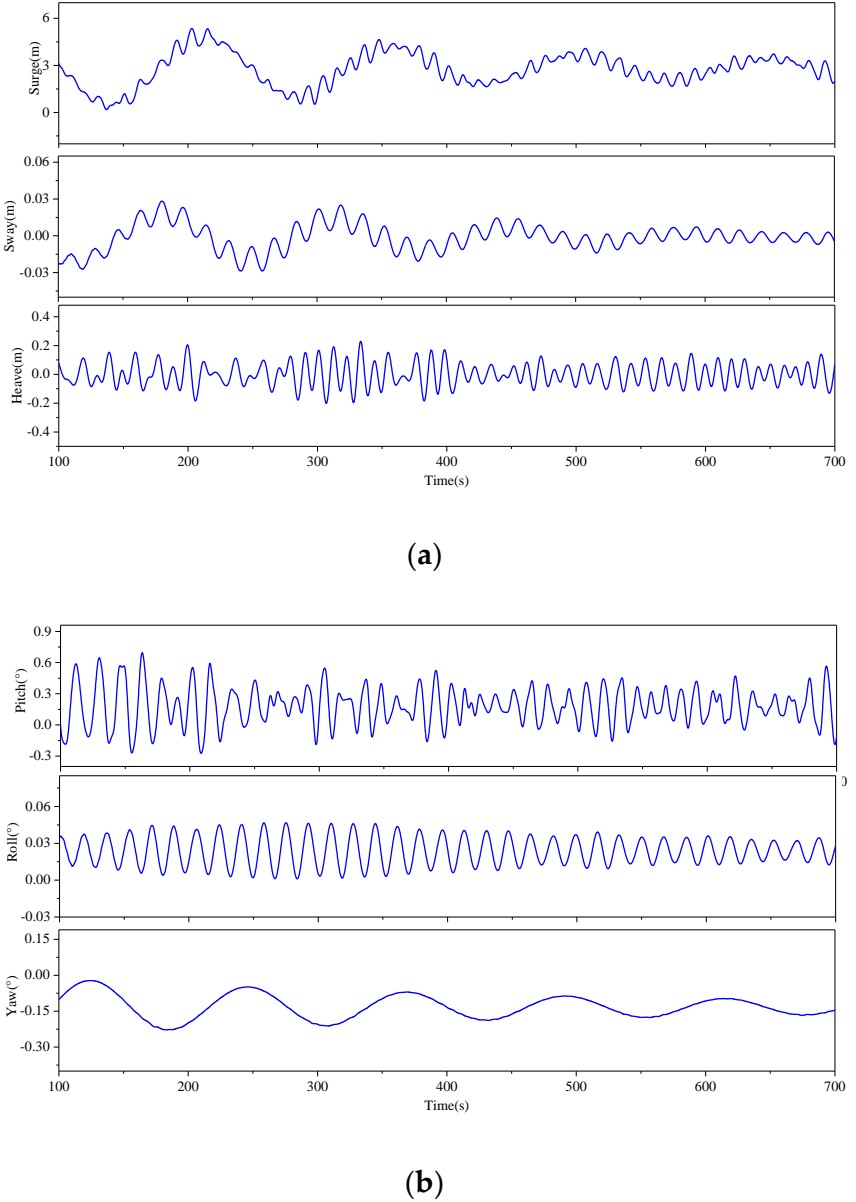

**(a)**

**(b)**

**Figure 12.** Time histories of cage motions under the uniform wind with irregular waves. (**a**) Translational motion. (**b**) Rotational motion.

Figure 13 illustrates the time series of blade pitch angles and wind turbine power output under the uniform wind with irregular waves condition. Due to the blade pitch angle variation tuned by the blade pitch controller, a mean generator power of 5,000 kW is output, with a small stand deviation of approximately 57 kW. The mean value of the blade

pitch angle is approximately 15° and there is only slight oscillation. This is attributed to the stability of the cage pitch motion, as shown in Figure 12.

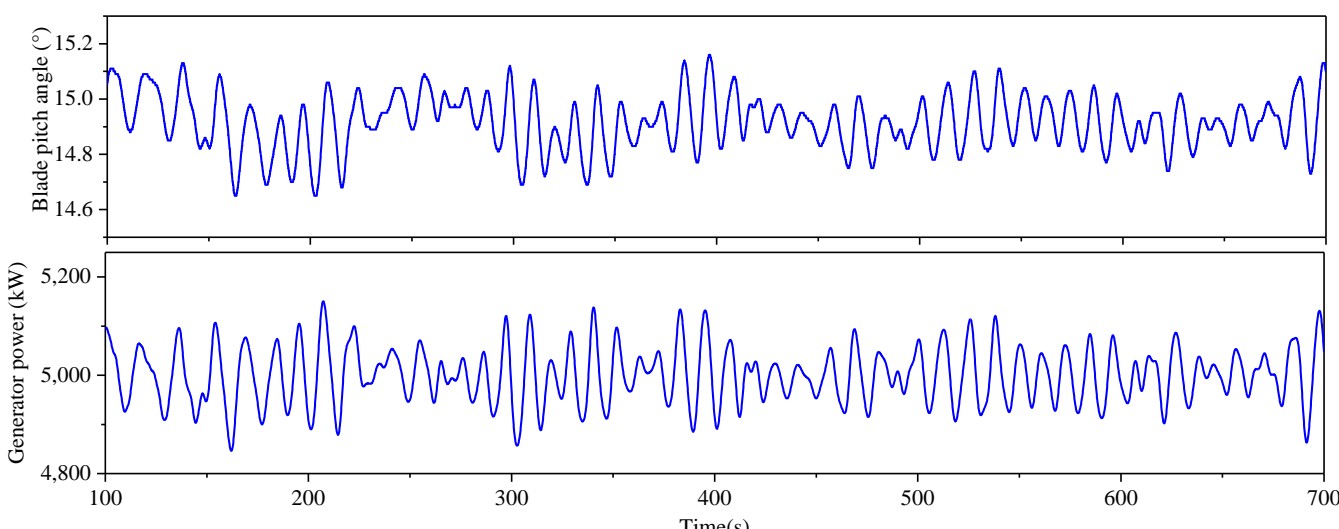

**Figure 13.** Time histories of pitch angles and power output of the wind turbine under the uniform wind with irregular wave.

### 5.3. Turbulent Wind and Irregular Wave Test

In this paper, the Norway 5 site is chosen as a representative site for checking the dynamic performance of the integrated system under the turbulent wind and irregular wave conditions, as shown in Figure 14 [19]. A generic water depth of 200 m is applied for this site.

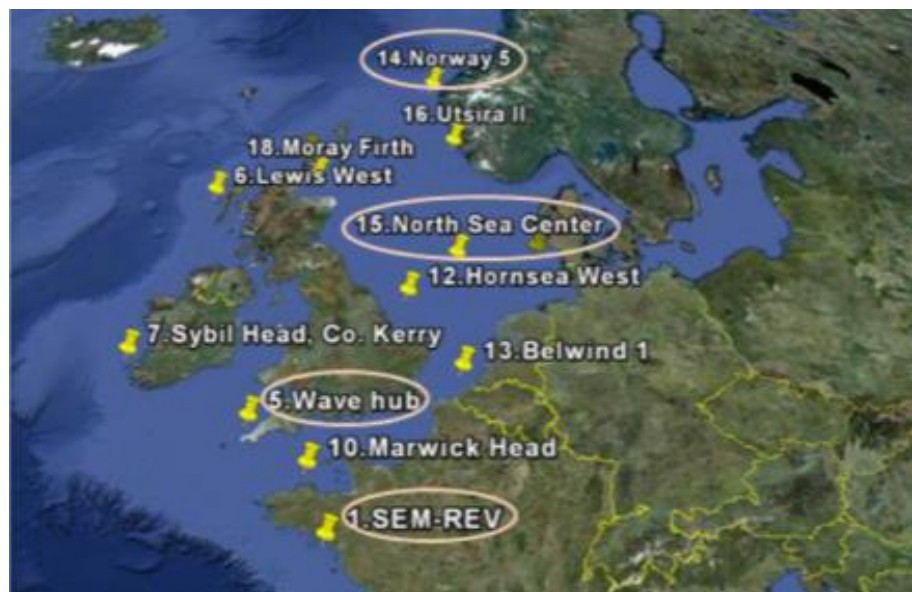

**Figure 14.** Location of the Norway 5 site [19]. Reprinted/adapted with permission from Ref. [19]. 2014, 20.19. Wang, Q.

The wind and wave data at the site have been fitted with analytical joint distributions by Li [44]. Therefore, it is possible to adopt the following procedure to determine operating loads cases [19]: (1) a mean wind speed at hub height is firstly selected, and then the mean wind speed is transformed to the reference height of 10 m via a power law. (2) The

conditional distribution of significant wave height for the given mean wind speed is used, and the most probable value of significant wave height is chosen. (3) The conditional distribution of the wave peak period for a given significant wave height and mean wind speed is used, and the most probable value of the wave peak period is determined.

Four load cases, including three operating cases and one parked case, are selected according to the above procedure [19], as listed in Table 8. Cases 1–3 correspond to the below-rated wind speed, rated wind speed and above-rated wind speed, respectively. Case 4 represents an extreme condition, which is carried out to examine the extreme cage motions. The cage horizontal offset should have a limitation to avoid vertical forces on the anchor, thus, an allowable offset of around 10% of the water depth [45] (that is, 20 m) is adopted in this paper.

**Table 8.** Environmental conditions for simulations.

| Case | Mean Wind Speed (m/s) | Significant Wave Height (m) | Wave Peak Period (s) | Turbulence Intensity (%) | State |
|------|------|------|------|------|------|
| 1 | 8 | 2 | 10.3 | 17 | Operating |
| 2 | 11.4 | 2.5 | 10.2 | 15 | Operatng |
| 3 | 18 | 4.1 | 10.5 | 13 | Operatng |
| 4 | 40 | 15.6 | 14.5 | 11 | Parked |

A normal turbulence model is applied for generating wind files for operating cases, while an extreme wind model is for the parked condition. The operating cases are the general power production cases with rotating blades and an active controller. However, all the blades are pitched to feather, and the wind turbine is shut down to avoid damage in extreme conditions; therefore, the wind turbine is parked. The Kaimal turbulence model is used to generate the wind condition. The extreme environment condition (parked) is obtained by the contour surface method with a return period of 50 years [44]. The total simulation time is 4000 s, and the results corresponding to the first 400 s are omitted.

5.3.1. Time Domain Analysis

For brevity of this paper, only the time histories of responses under cases 3 and 4 are provided as representatives, as shown in Figures 15 and 16. TwrBsMy and RootMy denote the tower-base fore-aft bending moment and the blade-root out-of-plane bending moment, respectively. Additionally, the time histories of wave elevation and horizontal wind speed are also plotted in the figures.

From the figures, it is found that the cage motions obviously oscillate in time due to the action of turbulent wind and irregular waves, especially the surge motion with a non-zero mean value. In particular, under the extreme load case, with the wind turbine parked and the blade pitched to the feather, the cage motion oscillation is much different from the operating case, because there is no damping from the controller and the oscillation is wave-dominated. Additionally, in the extreme condition, the maximum surge offset reaches up approximately 13 m, which is almost 65% of the allowable offset (20 m). It means that the stiffness of the mooring lines is just enough for the fishing cage.

The high oscillation In the curves of the blade pitch angle under condition 3 Is observed to maintain the rated generator power output of 5 MW. This indicates that the blade pitch controller is operating properly, in this case. However, there are still some places on the power curve below the rated value 5 MW, indicating that the wind speed on the rotor is dropped below the rated wind speed (11.4 m/s). Under the action of nonlinear aerodynamics, the tower-base fore-aft bending moment and blade-root out-of-plane bending moment oscillate significantly.

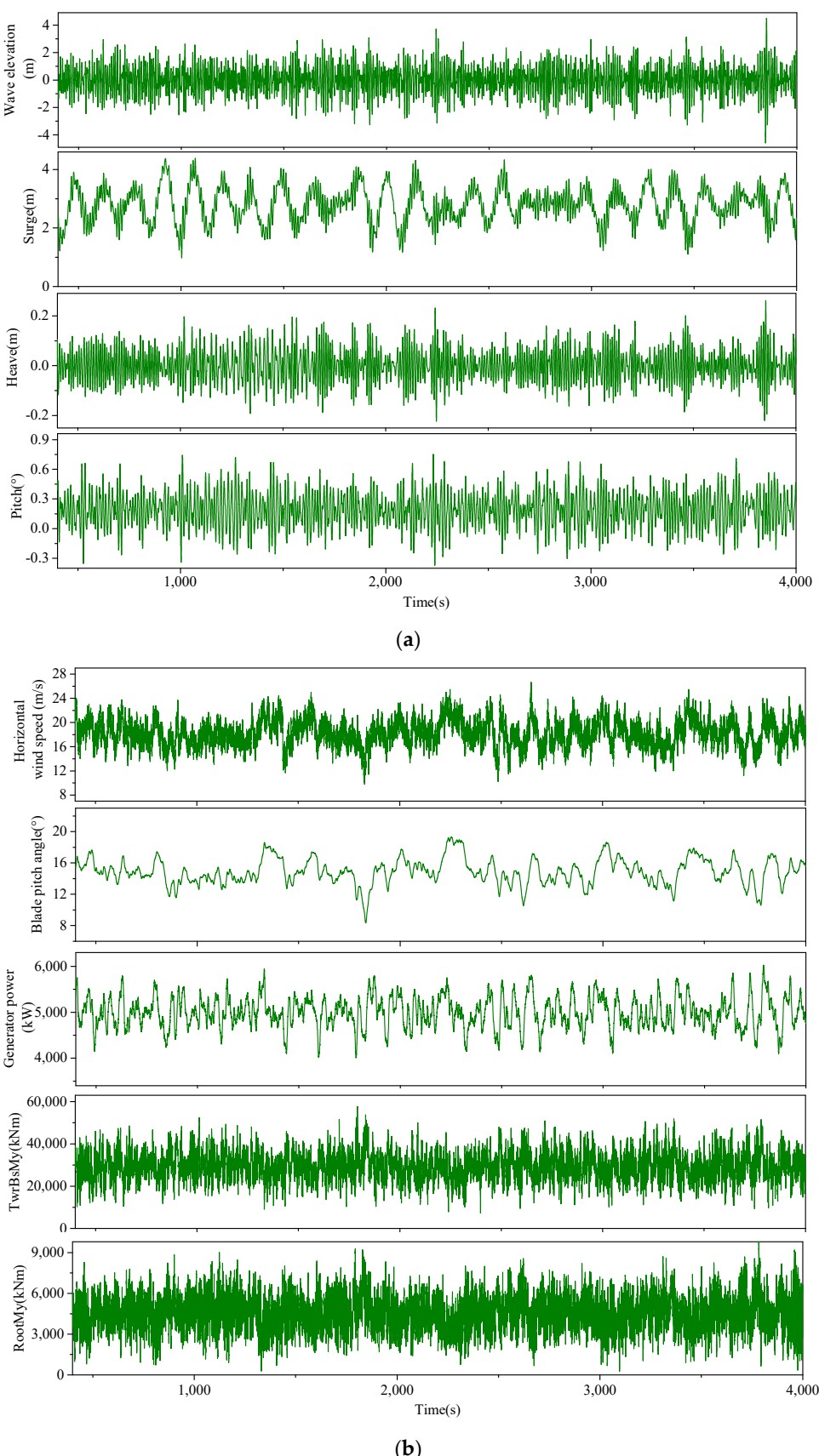

**Figure 15.** Time histories of responses of the cage and wind turbine under condition 3. (**a**) Cage responses. (**b**) Wind turbine responses.

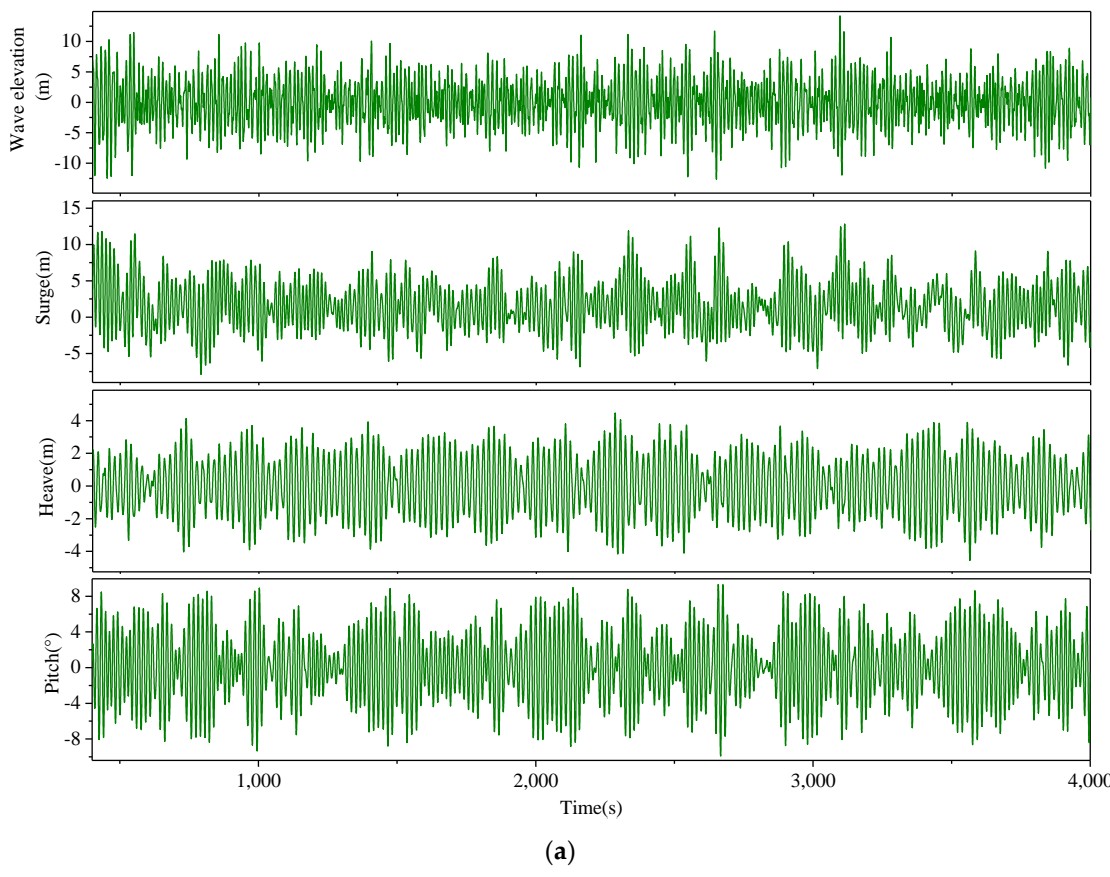

(**a**)

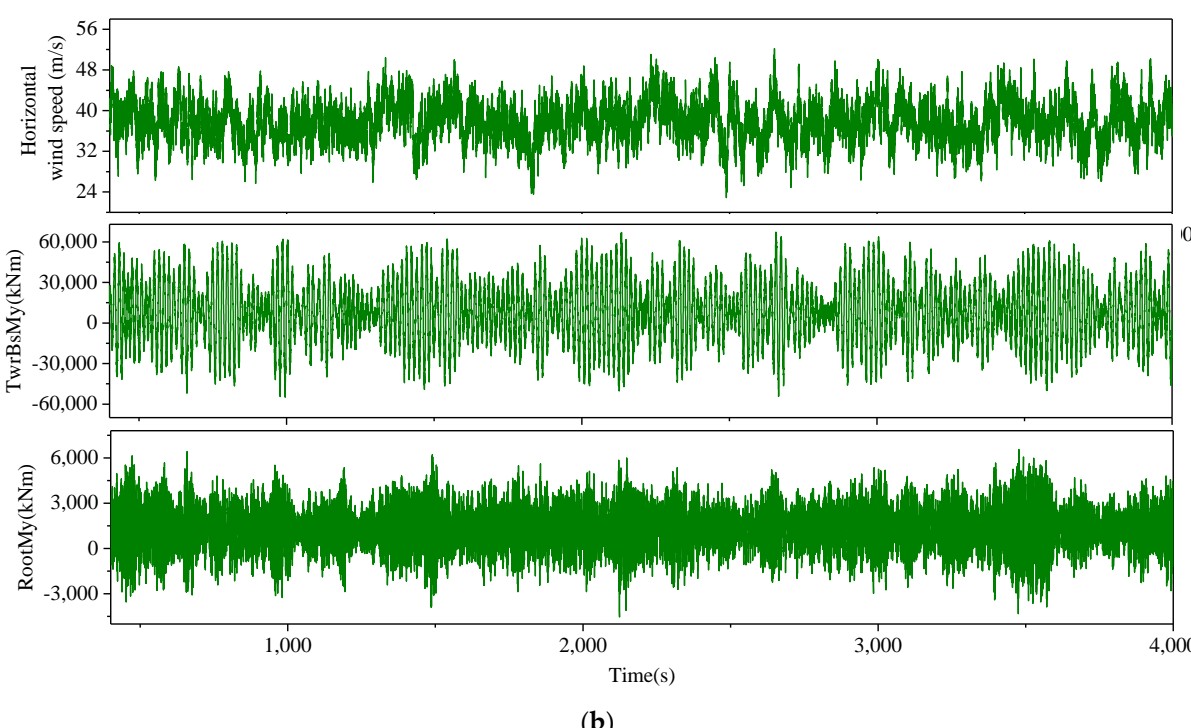

(**b**)

**Figure 16.** Time histories of responses of the cage and wind turbine under condition 4. (**a**) Cage responses. (**b**) Wind turbine responses.

Several statistical properties, including mean, absolute maximum (Abs Max) and standard deviation of the fishing cage responses (surge, heave and pitch motions) and wind turbine responses (blade pitch angle, generator power, tower-base fore-aft bending moment and blade–root out-of-plane bending moment) are calculated as evaluation indices, as shown in Figure 17. Standard deviations of responses are plotted as error bars on top of mean values.

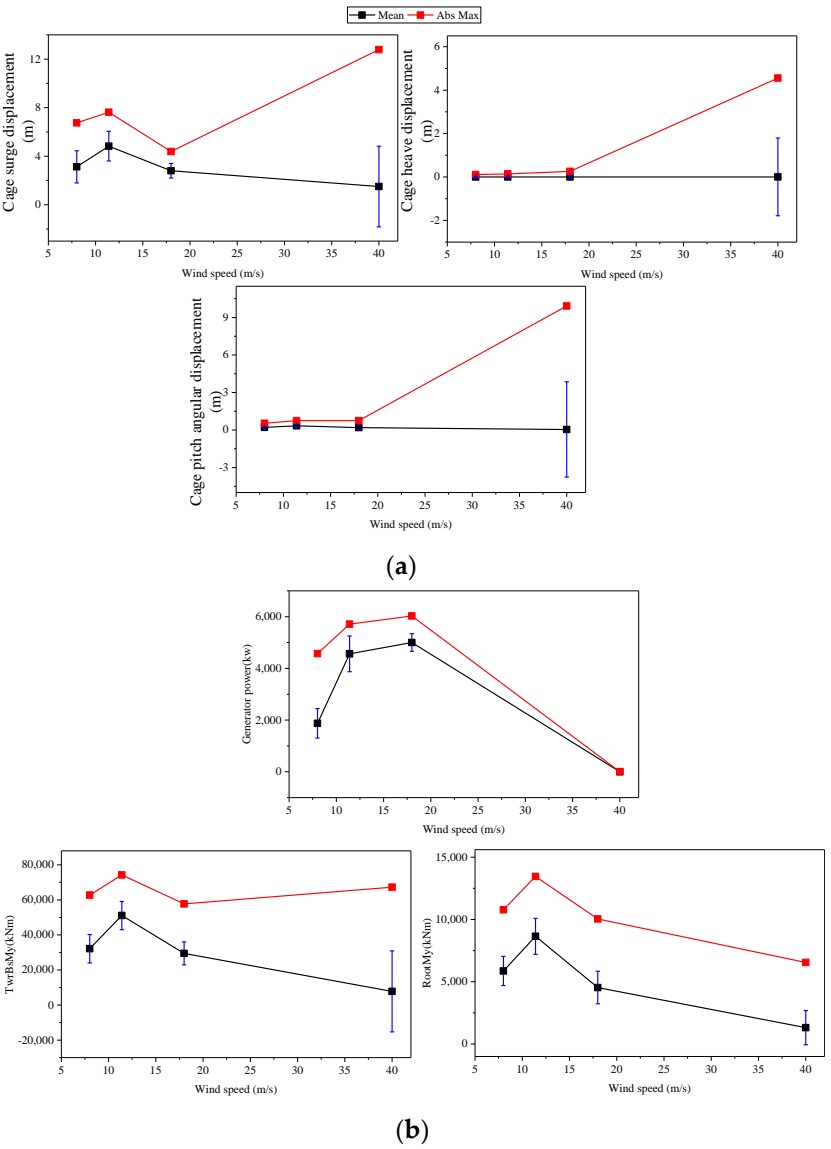

**Figure 17.** Statistical properties of the cage and wind turbine responses. (**a**) Cage responses. (**b**) Wind turbine responses.

For the statistical results of cage responses, it is seen that the heave and pitch motions exhibit similar patterns with almost zero mean values, while the surge motion dominates the cage response. The peak value of mean surge motions occurs at the rated wind speed condition. In addition, it is observed that the maximums and standard deviations of the surge, pitch and heave motions all occur in the extreme wind condition. For the statistical results of wind turbine responses, with the increase in wind speed, the generator power output increases, and no power is output at the parked condition. The standard deviation of generator power reaches the largest at the rated wind speed condition. This is expected because the wind speed fluctuates frequently around the rated value. The mean and maximum values of the blade-root out-of-plane bending moment show a similar pattern.

Specifically, the peaks of the mean and maximum blade-root bending moments occur at the rated wind speed condition, while the lowest values appear at the extreme condition. Moreover, the mean and maximum tower-base fore-aft bending moments have peak values at the rated wind speed condition, while the largest standard deviation occurs in the extreme condition.

### 5.3.2. Frequency Domain Analysis

In order to further investigate the effect of turbulent wind, a spectral analysis is performed for the cage and wind turbine responses. The smoothed spectra are shown in Figure 18. The spectra of wave elevation and horizontal wind speed are also plotted so that they can be used to compare each response spectrum and help to identify the source of the most energetic response.

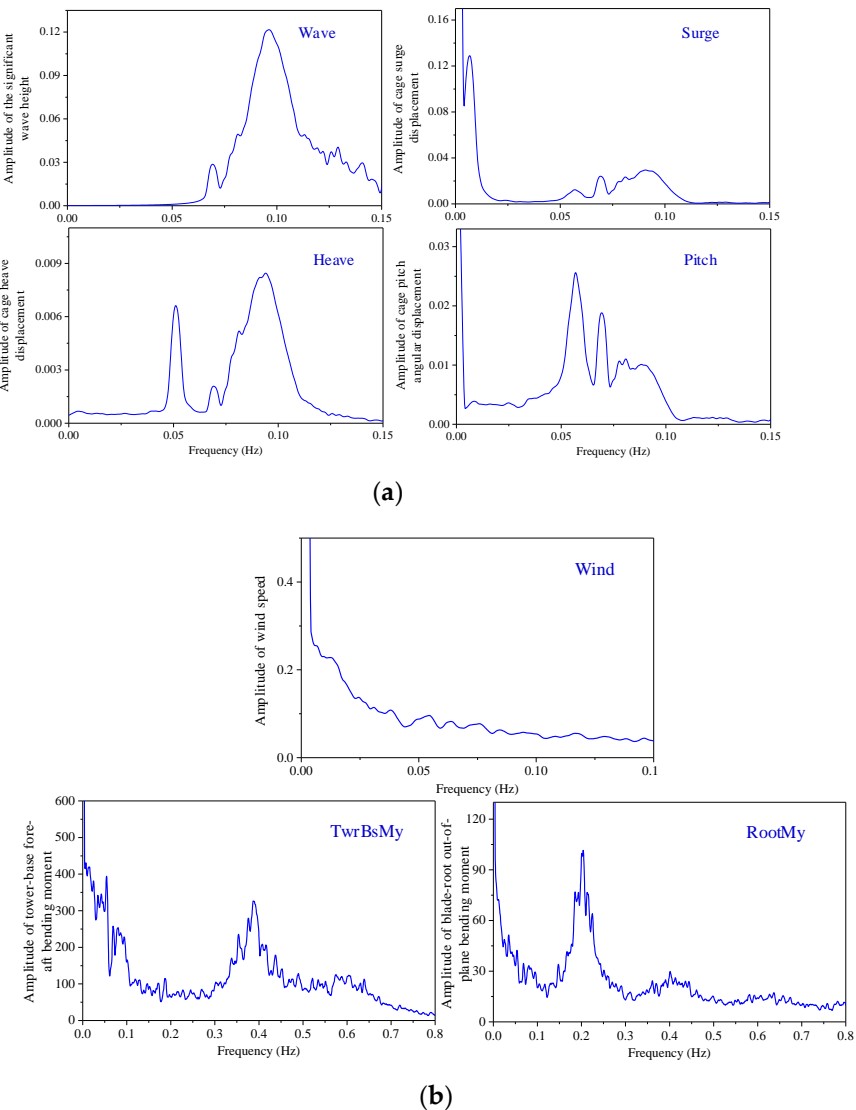

**Figure 18.** Smoothed spectra of the cage and wind turbine responses under condition 3. (**a**) Cage responses. (**b**) Wind turbine responses.

From Figure 18a, it is observed that the surge motion is dominated by the low-frequency responses due to the turbulent wind and surge resonant responses. A peak occurs at the surge resonant frequency, while the pitch motion and wave excitation also influence the surge motion. For the heave spectrum, two peaks occur at approximately 0.05 Hz and 0.1 Hz, which correspond to the heave natural frequency and wave natural

frequency component, respectively. The surge and pitch motions have no influence on the heave motion. For the pitch response, in addition to the low-frequency component of the turbulent wind, there are also obvious pitch resonance frequency and wave frequency components. Overall, the turbulent wind has some influence on the surge and pitch motions, but with no influence on the heave motion. The possible reason is that wind inclination is not considered in simulations, which means that the wind direction is always horizontal. Moreover, the wave frequency response generates some influence in all three DOFs.

From Figure 18b, it is observed that the tower–base fore-aft bending moment (TwrBsMy) is mostly affected by turbulent wind, in addition to the cage pitch motion, irregular waves and 2P-effect. Moreover, there is a small peak at approximately 0.6 Hz, which is possible due to the 3P-effect of the blades. For the blade–root out-of-plane bending moment (RootMy), it is obvious that the turbulent wind and 1P-effect have important effect on the blade response. Moreover, the 2P-effect and 3P-effect also contribute to the blade–root out-of-plane response.

### 5.4. Influence of Mooring Line Length

In order to study the influence of line length on the responses of the integrated system, two additional mooring line lengths (924 m and 968 m, which are 5% and 10% longer than the original length 880 m, respectively) are chosen, with an assumption of unchanged locations of fairleads and anchors. The cross-sectional area of each line is inversely proportional to its length so that the total mass of each line is unchanged. The time histories of responses for different line lengths under case 3 are illustrated in Figure 19. It is observed that for the cage responses, the increase in mooring line length mainly influences the surge motion, while the heave and pitch motions are slightly affected. The statistics of surge motion are illustrated in Figure 20. Note that the percentages in the figure indicate the reduction of the corresponding index relative to the result with respect to the 880-m-long line. The positive and negative values indicate increase and decrease, respectively. For example, 404.8% indicates that the maximum surge motion with a 924-m-long line is 404.8% larger than that with an 880-m-long line. It is seen that with the increase in line length, the surge motion becomes remarkable.

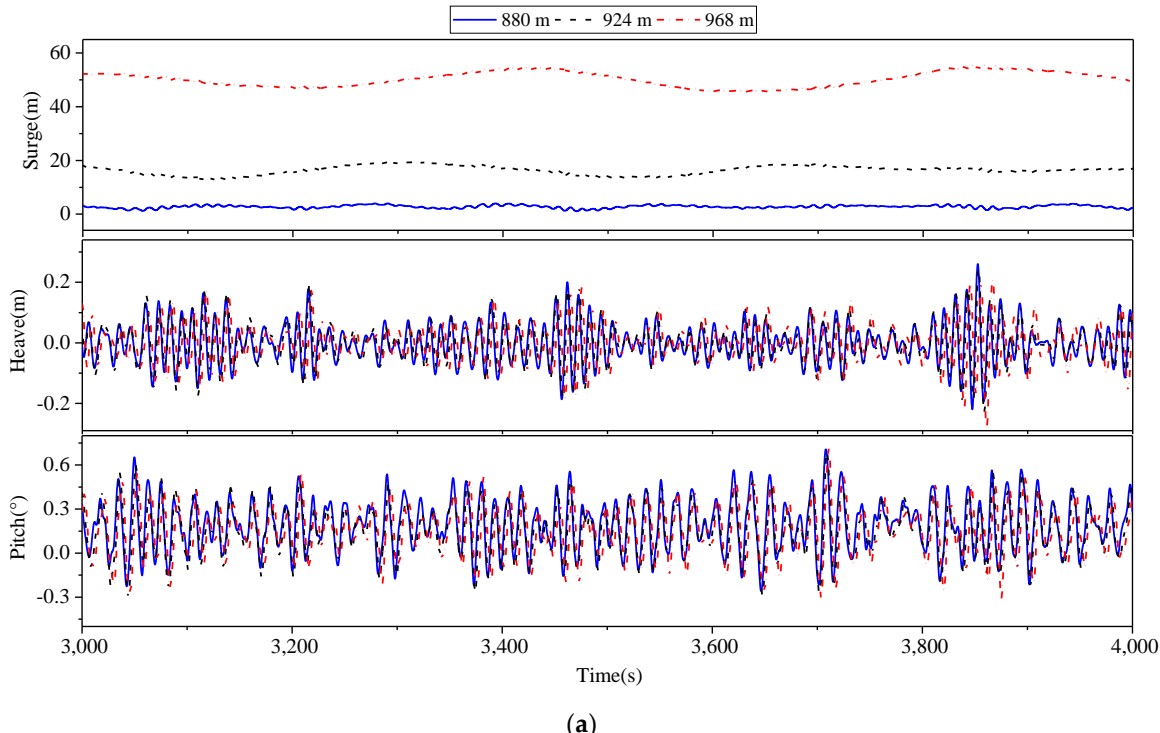

(**a**)

**Figure 19.** *Cont.*

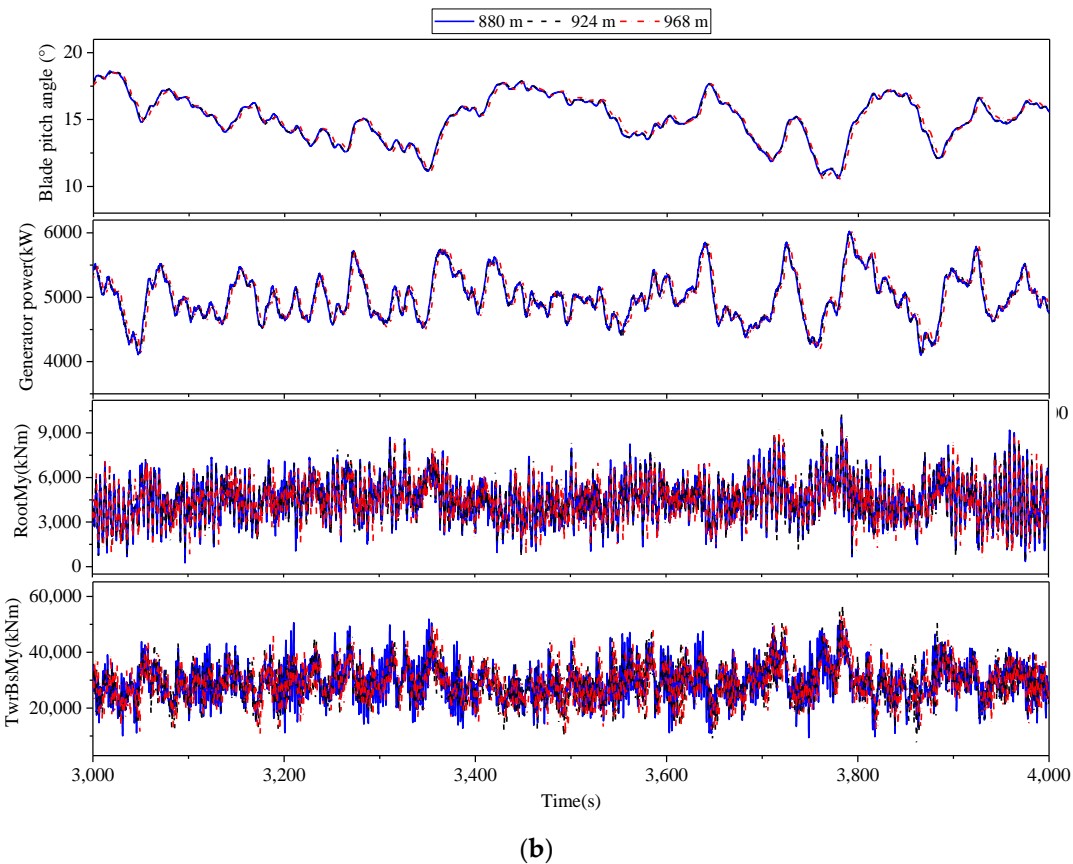

**(b)**

**Figure 19.** Comparison of time histories of the fishing cage and wind turbine responses for three mooring line lengths under case 3. (**a**) Cage responses. (**b**) Wind turbine responses.

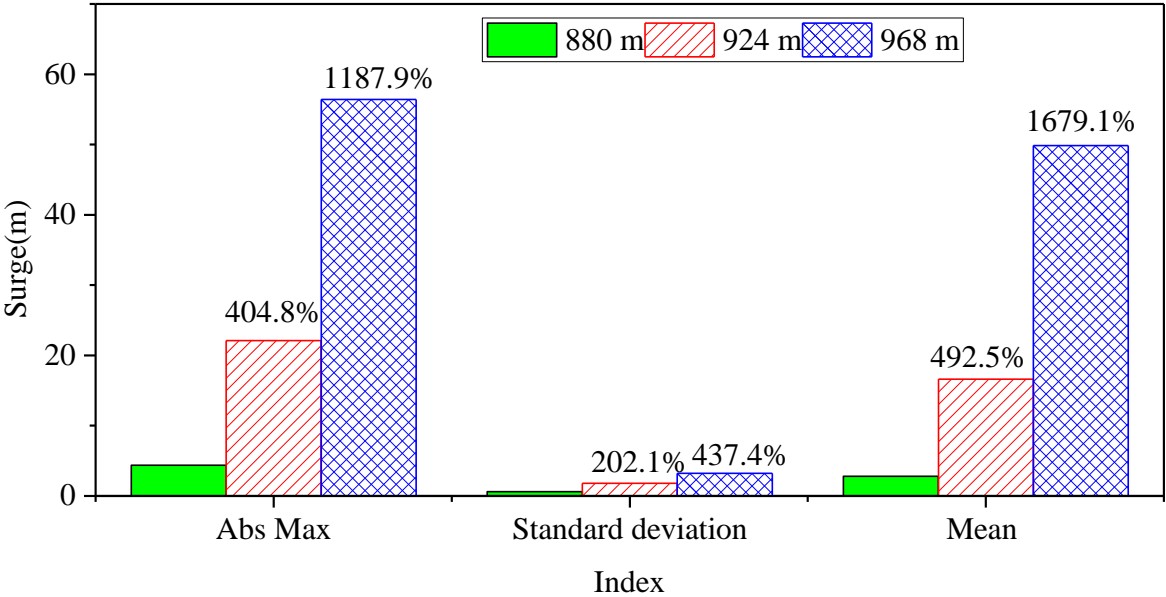

**Figure 20.** Statistics of surge responses for three line lengths.

As for the response of the wind turbine, although the surge movement is markedly affected, the increasing line length has a slight impact on them, which is attributed to the function of the wind turbine controller. However, the response curves have a certain offset backward. This is due to the large surge motion caused by the significant increase in line

length, which delays the time for the wind to reach the rotor, thus, delaying the wind turbine responses.

Figure 21 shows the comparison of the upwind mooring line tension at the fairlead for three line lengths. It is found that with the increase in mooring line length, the maximum and mean line tensions reduce. This change in trend is opposite to that of the surge motion. The increase in line length has less effect on the standard deviation of line tension, which is related to the fact that the increasing line length has little influence on cage pitch and heave motions, as shown in Figure 19a. In addition, it is seen that a further increase in line length does not further significantly reduce the line tension. To be specific, when the line length increases by 5%, the maximum line tension decreases by 45.7%, while when the line length increases by 10%, the maximum line tension only decreases by 52.9%. The same change trend occurs in the mean line tension.

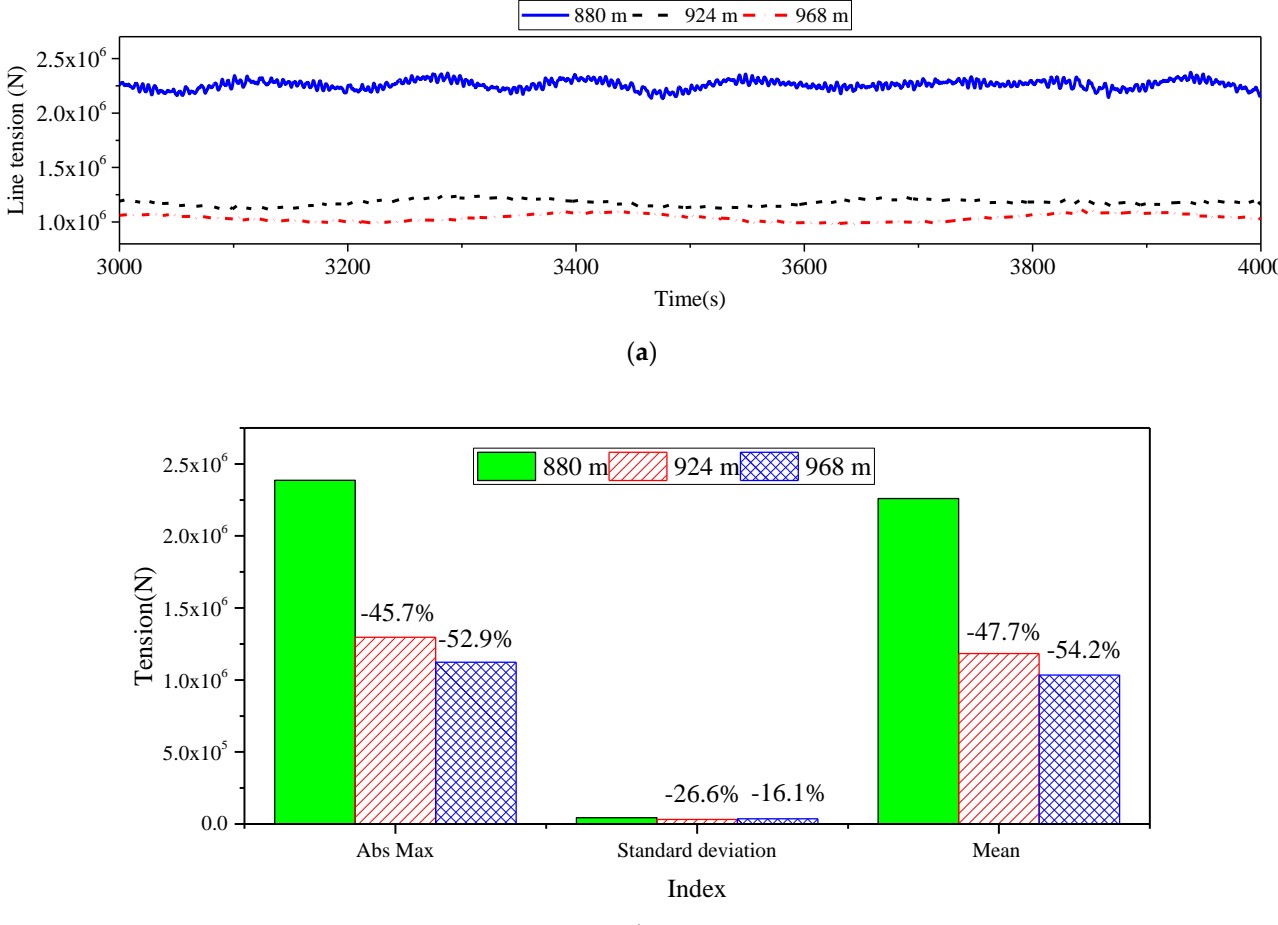

**Figure 21.** Comparison of upwind line tension at fairlead for three line lengths. (**a**) Time histories. (**b**) Statistics.

## 6. Conclusions

In this paper, a state-of-the-art concept of a floating offshore wind turbine integrated with a steel fishing cage is investigated. First, the structural configuration and dimensions of this integrated system are presented. Second, the dynamics model of the integrated system is established through FAST and AQWA. Specifically, the upper wind turbine system is modeled in FAST, while the lower fishing cage is modeled in AQWA. Information exchanges between the two codes. Third, for the designed integrated system, a blade-pitch generator-torque controller is applied. A coupled aero-hydro-elastic-servo model is then constructed. Finally, a series of simulations under selected load cases are performed to

explore the dynamic response behaviors of the integrated system. The influence of mooring line length is additionally studied. Key conclusions of this work are listed as follows.

(1) For the operating conditions, the rated wind speed condition is the most important condition for the integrated system because large pitch motion occurs, which, in turn, influences the generator power production. On the other hand, in the extreme wind conditions, surge motion becomes higher and exhibits significant oscillation.

(2) Overall, the turbulent wind has some influence on the surge and pitch motions, but with no influence on the heave motion. The wave frequency component can be seen in the surge, pitch and heave responses of the fishing cage, especially in the heave response. Additionally, the surge and pitch motions have no influence on the heave motion.

(3) The tower–base fore-aft bending moment is mostly affected by turbulent wind, in addition to the cage pitch motion, irregular waves, and blade rotation effect. For the blade–root out-of-plane bending moment, the turbulent wind and 1P-effect have an important effect on the blade response.

(4) At the above-rated condition, compared with heave and pitch motions, the cage surge motion is more affected by the increase in mooring line length. The wind turbine responses are slightly influenced by the increasing line length, but exhibit delay caused by the large surge motion. In addition, with the increase in mooring line length, the maximum and mean line tensions reduce, but the standard deviation of line tension is less affected. A further increase in line length does not further remarkably reduce the line tension.

It must be pointed out that the focus of this paper is to provide an idea of conceptual design, modeling and simulation analysis for the integrated wind turbine-fishing cage system. More detailed structural design optimizations, strength checks and experimental tests for the integrated system need to be further carried out in future.

**Author Contributions:** C.Z.: Simulation, Data curation, Writing-original draft, Writing-review & editing; J.X.: Investigation, Data curation; J.S.: Information retrieval, Software; A.L.: Simulation; M.C.: Funding acquisition; H.L.: Supervision, Methodology; C.G.: Data curation; S.X.: Supervision, Methodology, Writing-original draft. All authors have read and agreed to the published version of the manuscript.

**Funding:** This paper was funded by Key R & D program of Shandong Province demonstration project of deep sea aquaculture technology (No.: 2021SFGC0701); National Natural Science Foundation of China (No.: 41976194), Project of Science and Technology Research Program of Chongqing Education Commission of China (No.: KJQN202101133), and Scientific Research Foundation of Chongqing University of Technology (No.: 2020ZDZ023).

**Data Availability Statement:** The data is in the article.

**Conflicts of Interest:** The authors declare no conflict of interest.

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
