# Peer review of "Preliminary Study on an Integrated System Composed of a Floating Offshore Wind Turbine and an Octagonal Fishing Cage"

_jmse, doi:10.3390/jmse10101526_

Round 1
Reviewer 1 Report
Abstract: English corrections:
Page 1, Line 4: is designed to matches
Page 1, Line 9: at the operating conditions
Page 1, Line 10: various scenarios
Suggestion: Some numerical results may be useful in the abstract.
Introduction:
Can you add some existing cases being used with fishing cage/s for power generation offshore?
What is the existing installed capacity of such system/s globally?
Suggestion:
Please add a section describing the wind speed and wave intensities and currents surrounding the location considered for the present case study.
If possible, please add the wind power curve of the wind turbine.
Have you used the measured wind speed and direction data?
Please try to include the following papers in the literature part:
Wind Power Resources Assessment at ten Different Locations using Wind Measurements at Five Heights, Environmental Progress & Sustainable Energy Journal, https://doi.org/10.1002/ep.13853
Energy harvesting from passive oscillation of inverted foil, Physics of Fluids 33(7), 075111, https://doi.org/10.1063/5.0056567
Reviewer 2 Report
Dear Authors, thank you for the paper, which is very interesting, presenting a quite innovative technology: a concept of a floating offshore wind turbine integrated with a steel fishing cage. I have few comment. The Authors seem to never consider some recent work on laboratory experiments of floating wind turbine, in particular for effects due to rotating blades or vibratory loads. The structural model and the dynamic modelling are accurately described, however it is not clear how the environmental loads have been generated. Try to better define the hydrodynamics of the heave plate. Why the Authors used only irregular waves? Normally, in preliminary study, regular wave conditions are firstly required, due to the need of clarify the structural response, avoiding the "loss of control" of the results. It is quite difficult, in such kind of devices, to hold off the strong non-linear effects. Please, in figure 15 and 16 explicit the y-axis.Author Response
Please see the attachment

Reviewer 3 Report
This paper an idea of integrating a fishing cage into an offshore floating wind turbine. FAST and AQWA analyses have been performed. I appreciate the new idea, however, the main concern about the paper is not on the methodology and analysis which can be found in other literature, but more on the implementation of the idea.
Personally, I’m very sceptical about the idea. Having an offshore wind turbine is a million project while hatching fish can be done in any other region, but not with the floating wind turbine system while the investment and risk management are too high. Unless the author can convince me in terms of feasibility in technical, financial, risk, maintenance, parasitic effects, ROI etc. Otherwise, it is not worth for a scientific publication based on the authors' imagination that is feasible but not practical.
Parasitic effects of the fishing cage are a crucial problem, where the dynamic forces of the fishes, and growth of shell causing an additional weight to the turbine system need to be addressed, but not solely on the mooring, floating, motion, etc.
Have the author consider the depth of the sea, type of fish, coverage area, noise to the fish, and biological factors on the health of the fish. Or it is just a cage without fish inside?
Is there any solid data about fishes like to accumulate near the floating platform?
I would suggest the authors do not simply include or integrate an idea with others.
The content is mainly focused on the OFWT system, theoretical idea and calculation, where the practical content is missing, the author should have done some preliminary tests on the small model system to prove the concept.
In addition, the methodology and analysis performed are similar to other literature that just added the fish cage and fishnet, but yet to address the main concern of the topic
The hydrodynamic of fishes in the cage is not taken into consideration. This paper should be only published with a solid practical study even a small-scale model of up to 1kW turbine in shallow water.
Round 2
Reviewer 3 Report
I appreciate the effort done by the authors in revising the paper. However, the reviewer is not convinced by the responses given, where many issues raised are not addressed well. I'm very sorry that I have to reject the paper.